# Deep Impact: Shifts of Native Cultivable Microbial Communities on Fresh Lettuce after Treatment with Plasma-Treated Water

**DOI:** 10.3390/foods13020282

**Published:** 2024-01-16

**Authors:** Hauke Winter, Robert Wagner, Jörg Ehlbeck, Tim Urich, Uta Schnabel

**Affiliations:** 1Leibniz Institute for Plasma Science and Technology (INP), Felix-Hausdorff-Strasse 2, 17489 Greifswald, Germany; hauke.winter@uni-greifswald.de (H.W.); robert.wagner@inp-greifswald.de (R.W.); ehlbeck@inp-greifswald.de (J.E.); 2Institute of Microbiology, Center for Functional Genomics of Microbes, University of Greifswald, 17489 Greifswald, Germany; tim.urich@uni-greifswald.de

**Keywords:** lettuce, microbial community, physical plasma, microwave-driven plasma, Gram-positive bacteria, Gram-negative bacteria, yeasts, MALDI-ToF, microorganisms

## Abstract

Foods consumed raw, such as lettuce, can host food-borne human-pathogenic bacteria. In the worst-case, these diseases cause to death. To limit illness and industrial losses, one innovative sanitation method is non-thermal plasma, which offers an extremely efficient reduction of living microbial biomass. Unfortunately, the total viable count (TVC), one of the most common methods for quantifying antimicrobial effects, provides no detailed insights into the composition of the surviving microbial community after treatment. To address this information gap, different special agars were used to investigate the reduction efficiency of plasma-treated water (PTW) on different native cultivable microorganisms. All tested cultivable microbial groups were reduced using PTW. Gram-negative bacteria showed a reduction of 3.81 log_10_, and Gram-positive bacteria showed a reduction of 3.49 log_10_. Fungi were reduced by 3.89 log_10_. These results were further validated using a live/dead assay. MALDI-ToF (matrix-assisted laser-desorption-ionization time-of-flight)-based determination was used for a diversified overview. The results demonstrated that Gram-negative bacteria were strongly reduced. Interestingly, Gram-positive bacteria and fungi were reduced by nearly equal amounts, but could still recover from PTW treatment. MALDI-ToF mainly identified *Pseudomonas* spp. and groups of *Bacillus* on the tested lettuce. These results indicate that the PTW treatment could efficiently achieve a ubiquitous, spectrum-wide reduction of microbial life.

## 1. Introduction

Microbial communities are highly adapted and regulated to their environment so that some communities can exist only in specific areas [1]. The entirety of a microbial life and its genomes (bacteria, archaea, mold and yeast eukaryotes, and viruses) at a given location and time is described using the term microbiome [2,3,4]. 

Likewise, plants also create a specific environment for microbial communities [5]. Therefore, their microbiomes are particularly diverse and offer many ecological niches that are occupied with a wide variety of organisms [5,6]. Plants have plant-specific communities, which could be influenced by habitat-specific shifts [7]. Plants host a large variety of microbes on and inside their tissues in a lifestyle range from mutualistic to communalistic and pathogenic [8,9,10]. Unfortunately, the interaction between a plant’s microbiome and its fitness is currently unknown in most cases [11]. 

The phyllosphere is a primary interface of the plant with the environment [12]. The phyllosphere is home to bacteria and fungi that interact with the plant and together form a holobiont [13]. The term ‘holobiont’ describes the whole living thing being of a biological system (eukaryotic host together with a plurality of microorganisms (Figure 1)) [14].

Lettuce-like endive (*Cichorium endivia* L.) and other leafy greens are very popular plant-based foods and can be sources of vitamins and nutrients for the human diet [15,16]. Because lettuce is usually consumed raw, a high degree of food safety must be achieved without damaging the plant tissue too much in order for the raw product to be purchased by consumers [16,17].

Commonly, the phyllosphere of lettuce can host 10^5^–10^7^ colony-forming units (CFU) per gram of lettuce fresh weight, and it is where bacteria are the most abundant colonizers [15,18]. In the 1990s alone, 12% of illnesses were food associated [19]. Examples of this are infections with *Listeria monocytogenes* and *Escherichia coli*, which can be found on lettuce too [18,20,21].

In addition, members of the native microbial community can also cause food to spoil. The yearly harvest and market loss of vegetables in Germany alone is about 1.1 million tons, with losses of 30% considered normal [19]. Furthermore, there are specific spoilage-associated bacteria like *Pseudomonadaceae* and *Enterobacteriaceae*, which decrease the shelf life of packed lettuce [22,23]. Therefore, reducing the total microbial count on food is the method of choice [24,25,26].

There is a high variance of sanitation methods used to process a clean and safe lettuce: conventional methods use pure water, or chlorine dioxide, and ultraviolet rays are used as additives to clean and cycle the wash water, as clean water is vital to reduce the bacterial load on vegetables [27,28,29].

An innovative and alternative method for food sanitation may be the use of non-thermal atmospheric pressure plasma [30,31]. In general, physical plasma describes a gas-particle mixture of ions, free electrons, and many neutral atoms or molecules which fulfill the roles of charge-free carriers [32,33]. The usage of non-thermal but warm plasma in direct application is limited to heat-resistant materials and cannot be used directly for lettuce sanitation [30]. However, indirect sanitation of lettuce via physical plasma is possible using plasma-treated water instead of pure tap water.

The MidiPLexc is a microwave-driven plasma torch, which works with compressed air as the working gas [31]. This plasma torch generates a complex cocktail of reactive species in the plasma-processed air (PPA) [34]. The PPA components in turn react with other mediums like water and generate plasma-treated water (PTW). The MidiPLexc mainly generates nitric oxide (NO*), nitrogen dioxide (NO_2_*), and probably also hydrogen peroxide (H_2_O_2_), as well as other metastable radical nitrite/oxide species (RNOS) [19]. For air at atmospheric pressure, the maximum nitric oxide concentration of 5.2% is achieved at about 3500 K [35]. The maximum concentration of NO_2_ is achieved at about the same temperature and is more than three orders lower. Other nitrogen oxides are negligible. At this temperature only, atomic oxygen is available in high concentrations of about 12%, which is similar to its molecular concentration. Only nitric acid (HNO_3_) could thus far be verified in the PTW [33]. Tap water includes nitrite, which is a stable reservoir that can be converted to NO* and other bioactive nitrogen species [36,37]. Several studies showed the reduction of the total viable count (TVC) on food after PTW treatment [19,30,31,33,38].

However, there are still many issues that require further investigation. The aim of this work is to provide further insight into the structure of the cultivable lettuce microbiome and the impact of PTW on it. Our core hypothesis is that the treatment with physical plasma will have a reductive effect on the native lettuce microbiome. In this study, we use proliferation assay (TVC), live/dead assay and matrix-assisted-laser-desorption-ionization time-of-flight (MALDI-ToF) analysis of randomly selected colonies at different time points to analyze the effect of PTW on the microbial communities of lettuce, especially endive (*Cichorium endivia* L.). To ensure that no higher nitrate concentrations accumulate in the lettuce during PTW treatment, the anion and cation balance of the lettuce was analyzed.

## 2. Materials and Methods

### 2.1. PTW Generation with Microwave Discharge

The MidiPLexc is an in-house-invented microwave-operated plasma system that operates under atmospheric pressure (2).

It is a further development of the in-house invention MiniMIP [39], which was operated with compressed air instead of noble gases (MiniMIP = argon, helium). Its use is for indirect plasma treatment of materials that would not survive direct plasma treatment due to the high temperatures involved. The plasma treatment is called “indirect” because the plasma effluent does not directly hit the medium to be treated, but only the compressed air is directly treated with the effluent. The resulting chemical compounds, which are sometimes reactive, then encounter the sample to be treated. The MidiPLexc (Figure 2) was operated with a forward power of 50 W at 2.45 GHz. The reflection from the plasma source was below 2.0 W. The gas flow amounted to 5.0 SLM and had a set point of 1.5 SLM. In parallel, the plasma aggregate was constantly cooled with compressed air via another inflow hose. The gas temperature of the effluent was about 1000 K. The distance between the plasma source and the water surface was approx. 23.5 cm. Plasma-treated water (PTW) was generated under constant mixing (300 rpm) with a PCE-MSR 100 magnetic stirrer (PCE Instruments; Meschede, Germany. For 90 min, 100 mL of tap water was exposed to constantly generated PPA. After the 90 min treatment, pH and conductivity were measured to validate the freshly generated PTW. The PTW was stored at 4 °C in the fridge until use.

### 2.2. PTW Treatment of Bite-Sized Endive Pieces

For treatments, 25 g of lettuce pieces were put into a 1000 mL bottle. In parallel to each PTW treatment, a control was performed with tap water (TW) and untreated lettuce as well. One hundred mL of PTW was added to the lettuce pieces, and the treatment time started immediately. The bottle was shaken throughout at 250 rpm using Ecotron laboratoy shaker (Infors HT; Bottmingen, Switzerland). After 1, 4, 7, and 10 min of PTW treatment, 5 g of lettuce and 20 mL of the PTW were removed. During the TW treatment, a sample was only removed after 10 min. The removed PTW was discarded, and the lettuce sample was blended with 20 mL of phosphate-buffered solution (PBS, pH 7.2 after Sorensen) for 30 s. Finally, 1 mL of the obtained supernatant was removed and used for further tests.

### 2.3. Lettuce Conditions

In this study, the lettuce studied was endive (*Cichorium endivia* L.). The endive was planted and harvested in May 2022 (Sand 1 Knick; between Jagsthausen and Heilbronn, Southwest Germany) on an open field. The harvested lettuce heads were stored cold and sent overnight at cool temperatures (5–8 °C) to Greifswald, Germany. In Greifswald, the lettuce heads were stored at 4 °C until use (max. for 1 week). For the experiments, the lettuce was cut into bite-sized pieces (~3 cm × 3 cm), which were finally used.

### 2.4. Proliferation Assay Based on Total Viable Count (CFU/mL)

For the first definitions of the treatment times, plate count agar (PCA) was used as a standard medium. After the treatment times had been defined, the impact of PTW on different groups of the microbial community was tested with special agars (Table 1).

After the PTW treatment (as well the untreated lettuce and the TW treatment) of the lettuce, the resulting 1 mL supernatant was blended and used for different assays. For the proliferation assay, 100 µL of it was diluted with 900 µL as the minimum required dilution (MRD; 8.5 g/L NaCl and 1.0 g/L tryptone). Thus, the sample was diluted stepwise up to 10^−5^ of the original concentration. The six dilutions 10^0^–10^−5^ were plated on the different agars using the Miles and Misra method with some adaptation [40] In brief, a 10 µL spot was spread using horizontal tilting of the agar plate. Subsequently, an incubation at room temperature (except EA, which was incubated at 37 °C) for 5 days took place. On days 2, 3, 4, and 5, the colonies on the agar plates were counted manually, and, finally, the CFU/mL was calculated. In addition, the unused PTW and TW, as well as the PBS were plated out as the control for contaminations.

### 2.5. Live/Dead Assay

For the detection of damaged cell walls, the ratio of two different fluorescent dyes (SYTO9 and propidium iodide (PI)) was measured. After PTW treatment, the ratio of SYTO9 (green)/PI (red), labeled G/R ratio, was detected using a LIVE/DEAD BacLight Bacterial Viability Kit (Thermo Fisher Scientific, Dreieich, Germany). The kit was used according to the instructions. For this, 9 µL of propidium iodide (PI) and 9 µL of SYTO9 mixture were added to 3 mL of ultrapure water. Then, 100 µL of the mixture was added to a 96-well plate followed by 100 µL of sample. Afterwards, the 96-well plate was incubated for 15 min in the dark at room temperature, while undergoing shaking at 80 rpm with the multi-functional orbital shaker PSU-20i (Biosan, Riga, Latvia). After incubation, the fluorescence signal of each well was detected using a Varioskan-Flash^®^ Enzyme-linked Immunosorbent Assays (ELISA) plate reader (Thermo scientific, Dreieich, Germany) at an excitation wavelength of 485 nm and an emission wavelength of 530 nm for SYTO9 and 630 nm for PI fluorescence. To calculate the ratio G/R, the green absorbance value was divided by the red absorbance value. Finally, the ratio G/R of the treated and control samples was put into a percentage ratio.

Liquid microbial samples were analyzed at different PTW-treatment times (0 s as control, 60 s, 240 s, 420 s, and 600 s). A G/R ratio was measured. Untreated samples (0 s) were used as 100% standard.

### 2.6. MALDI-ToF Analysis of Culturable Microbial Colonies

MALDI-ToF (matrix-associated-laser-deionization/ionization time-of-flight mass spectrometry) analysis was used as a tool to gather first insights into the cultivable microbial community composition on endive.

After each treatment time, 2–5 colonies on PCA were selected randomly and transferred to new PCA plates. Isolates were cultivated at room temperature. The isolate transfers were repeated several times and isolates were checked visually for purity. Some isolates did not grow after a few repeated transfers. Finally, 51 isolates were sent to the RIPAC-LABOR GmbH (Potsdam, Germany) for MALDI-ToF analyses (SHIMADZU AXIMA Assurance iDplus™). For microbial identification, the software AnagnosTec SARAMIS was used.

### 2.7. Ion Chromatography (IC) of the Lettuce Samples

All tested lettuce samples were freshly prepared and used on the same day (storage at 7 °C). Samples were only used once and were not reused.

For sample preparation, 5 g of lettuce was shredded and mixed with 50 mL of 70 °C heated water. After 15 min of incubation, the samples were filtered with folded qualitative filter paper (size 185 mm, slow filtration rate, particle retention 2–3 µm (VWR, Darmstadt, Germany)). As the next step, the pre-filtered samples were filtered with 0.2 µm filter (Filtropur S 0.2, Sarstedt AG & Co. KG, Nümbrecht, Germany). As the final step, the double-filtered samples were diluted 1:10 with ultrapure water (GenPure Pro Barnstead, Thermo Fisher Scientific, Dreieich, Germany).

Ion chromatographic (IC) measurements were performed to determine the anion (nitrate (NO_3_^−^), nitrite (NO_2_^−^), bromide (Br^−^), sulphate (SO_4_^2−^), chloride (Cl^−^), carbonate, phosphate (PO_4_^3−^)), and cation (sodium, magnesium, potassium, ammonium, calcium) concentrations in the distilled water and PTW. Measurements were performed with a Dionex ICS 6000 system (Thermo Scientific, Dreieich, Germany) equipped with a conductivity detector. The system was controlled using Dionex Chromeleon Version 7.1.3.1541. Ion separation was performed via an anion-exchange column (Dionex IonPac AS 18, Thermo Scientific, Dreieich, Germany), a guard column (Dionex IonPac AG 18, Thermo Scientific, Dreieich, Germany), and a cation-exchange column (Dionex IonPac CS 16) with another guard column (Dionex IonPac CG 16). The 23 mM KOH cartridge (Thermo Scientific, Dreieich, Germany) was used in combination with ultrapure water (GenPure Pro Barnstead, Thermo Fisher Scientific, Dreieich, Germany) as the eluent for the anion analysis. As eluent for cation analysis was used a combination of a 40 mM methane sulfonic acid (MSA) cartridge (Thermo Scientific, Dreieich, Germany) and ultrapure water. The injection of samples was performed using a partial-loop injection method. Volumes of 5 μL (anions) and 10 µL (cations) were used. Each lettuce sample was diluted 1:10, except for the distilled water, which was undiluted.

Thirty min was the runtime for each measurement and had an eluent flow of 0.25 mL/min (anions) and 0.36 mL/min (cations). The lettuce samples were stored in the autosampler at 10 °C (±5 °C).

The experiments were repeated twice with n = 3, resulting in n = 6. The identification and quantification of the different anions based on the Dionex 7-ion standard solution (Thermo Scientific, Dreieich, Germany). They were diluted with ultrapure water in samples of 1:1, 1:2, 1:4, 1:8, 1:16, 1:32, 1:64, and 1:128. Only the carbonate standard was treated different. All anions, except carbonate, were measured at the same eluent flow with 21 mM KOH. The carbonate was quantified with a 999 mg/L (±15 mg/L) standard (Sigma-Aldrich, Laramie, WY, USA) diluted in 50 mg/L steps from 500 to 150 mg/L with ultrapure water. The dilution steps were measured at 0.25 mL/min eluent flow and 23 mM KOH.

As a cation standard, a manually prepared mixed standard from magnesium, ammonium, potassium, and calcium was used(Sigma-Aldrich (Merck), Darmstadt, Germany). The cation standards had a concentration of 100 mg/L each. The mixed standard was then 1:1, 1:2, 1:4, 1:8, 1:16, and 1:32, diluted with ultrapure water, and measured with 40 mM MSA, and a 0.36 mL/min eluent flow. The sodium standard was prepared from sodium chloride ≥ 99%, Ph. Eur., USP (Carl Roth GmbH, Karlsruhe, Germany) solved in ultrapure water with a concentration of 100 mg/L and ratios of 1:1, 1:2, 1:4, 1:8, 1:16, and 1:32 diluted with ultrapure water. The sodium standard was then measured under the same conditions.

The concentrations of all measured ions were calculated using either the linear-regression function (all ions but ammonium) (1) or polynomial fitting (second-order polynom) (2) of the measured standards.
(1)$$c=\frac{(A\;\pm \;\;y_{intercept})\;}{x_{slope}}$$
(2)$$A=\alpha \cdot c^{2}+\beta \cdot c+\gamma$$
(3)$$c=-\frac{\sqrt{-4\;\alpha \gamma +4\alpha A+\beta^{2}}+\beta }{2\alpha }\;and\;\alpha \neq 0$$
(4)$$c=\frac{\sqrt{-4\alpha \gamma +4\;\alpha A+\beta ²}-\beta }{2\;\alpha }\;and\;\alpha \neq 0$$
(5)$$c=\frac{A-\gamma }{\beta }\;and\;\alpha =0\;and\;\beta \neq 0$$
where *A* is the integrated area under the conductivity peak over the retention time of the detected anion, *x_slope_* is the slope of the regression function, and *y_intercept_* is the interception of the *y*-axis of the regression function.

### 2.8. Statistical Tests for Differences in the TVC

A Kruskal–Wallis test with n = 9 and *α* = 0.05 was carried out using Origin Pro 2022b to determine whether a significant difference in the TVC was detected between the five treatments and the control for each agar. In addition, a Dunn’s post hoc analysis was carried out to determine which treatments had a significant difference. The *α* for the Dunn’s analysis had to be adjusted due to multiple tests. The Bonferroni method was chosen to adjust the *α*. Therefore, the total *α* of 0.05 was divided by the number of pairwise comparisons *k* (2–4).
(6)$$\alpha_{adjusted}=\frac{\alpha }{k}$$
(7)$$\mathrm{with}\;k=\frac{\#groups\;\cdot \;(\#groups-1)}{2}$$
(8)$$\mathrm{such}\;\mathrm{that}\;\alpha_{adjusted}=\frac{0.05}{\frac{6\times 5\;}{2}}\;\approx 0.0033333$$

## 3. Results

### 3.1. Proliferation Assay of PTW-Treated and TW-Treated Lettuce

To gain insight into the influence of PTW in comparison to pure tap water on the different parts of the native microbial communities on lettuce, different agars were used (Figure 3). The groups were represented as fastidious microorganisms (MOs) (PCA), non-fastidious MOs (MM), Gram-negative (MCA) and Gram-positive (CNA) bacteria, yeasts and fungi (S4), sporulating bacteria (GYEA), and coliforms at cultivation temperatures of room temperature (RT) (KVGL) and 37 °C (ENDO). Total-viable-cell counts (TVC; CFU/mL) were calculated.

#### 3.1.1. Proliferation Assay of Full Medium (PCA) and Shortage Medium (MM)

The concentration of TVC_PCA_controls_ increased until day 5 of incubation. The TVC_PCA_60s_ were decreased by 0.68 log_10_. Over the period of 5 days, the TVC_PCA_60s_ were nearly stable. The TVC_PCA_240s_ dropped in comparison to the TVC of shorter PTW-treatment times. After 5 days of incubation, the TVC_PCA_240s_ was comparable to day 2. The TVC_PCA_420s_ decreased more than the TVC_PCA_240s_, but increased slightly at longer incubation times. The TVC_PCA_600s_ (3.08 log_10_) was reduced to the same level as TVC_PCA_420s_ (3.08 log_10_). In contrast, the TVC_PCA_600s_TW_ was used as a counterexample to TVC_PCA_600s_. Therefore, the TVC_PCA_600s_TW_ (1.21 log_10_) was only decreased to the level of TVC_PCA_240s_ (1.56 log_10_). At all PTW-treatment times, a small increase in countable TVC was detectable with increasing incubation times.

Interestingly, the TVC_MM_ showed an analogous reduction over the different PTW-treatment times. The TVC_MM_controls_ was slightly higher than the TVC_PCA_controls_. The TVC_MM_60s_ after day 2 did not increase until day 5. The TVC_MM_240s_ was lower than TVC_MM_60s_. The TVC_MM_420s_ (3.07 log_10_) decreased similar to TVC_PCA_420s_ (3.11 log_10_). With a longer cultivation of up to 5 days, the TVC_MM_420s_ did not rise. For both the TVC_MM_420s_ and TVC_MM_600s_, similar reductions compared to PCA were obtained. Additionally, the TVC_MM_600s_TW_ resulted in a reduction (1.21 log_10_). However, this was much lower than the reduction of TVC_MM_600s_ (3.10 log_10_).

#### 3.1.2. TVC-Assay of Special Agars for Gram-Negative (MCA), Gram-Positive (CNA), and Yeast and Fungi (S4)

To check the three major groups of microorganisms (Gram-positive, Gram-negative, as well as yeasts and fungi), three different agars were used. For Gram-negative bacteria, MCA was used, while CNA was used for Gram-positive bacteria and S4 for yeasts and fungi.

The TVC_MCA_controls_ was stable at the incubation days 2 to 5. The TVC_CNA_controls_ were slightly lower than the TVC_MCA_controls_ on day 2. The TVC_CNA_controls_ rose at day 3 and maintained this concentration until day 5. The TVC_S4_controls_ were at the same level as the TVC_CNA_controls_.

The TVC_60s_ was stable for MCA after day 2 of incubation. TVC_CNA_ was at the same level on day 2. TVC_S4_ was slightly higher at day 2. Only for TVC_CNA_60s_, the CFU/mL rose with increasing days of incubation. The TVC_240s_ was decreased significantly on all three special agars with the PTW treatment (1.67 log_10_ (MCA); 2.31 log_10_ (CNA); and 1.59 log_10_ (S4)). The TVC_MCA_240s_ was relatively stable, while both the TVC_CNA_240s_ and TVC_S4_240s_ increased until day 5 of incubation. A 420 s PTW treatment reduced the TVC on all three agars significantly (3.86 (MCA); 3.55 (CNA); and 3.63 (S4)). Similar to the TVC_PCA_ and TVC_MM_, the TVC_MCA_, TVC_CNA,_ and TVC_S4_ were on the same agar species after a 420 s and 600 s PTW treatment. The TVC_MCA_600s_ nearly went below the detection limit. The TVC_CNA_600s_ was also close to the detection limit at day 2. However, with longer incubation times the TVC_CNA_600s_ rose. The TVC_S4_600s_ rose minimally over the incubation time.

The 600 s water treatment (1.48 log_10_ (MCA) and 1.58 log_10_ (S4)) achieved a reduction of the CFU/mL comparable to TVC_240s_ (1.67 (MCA) and 1.59 log_10_ (S4)) for two of three agars at day 2 after treatment. While TVC_MCA_600s_TW_ was stable, TVC_CNA_600s_TW_ increased, as well as TVC_S4_600s_TW_.

#### 3.1.3. Proliferation Assay of Special Agar for Sporulating Bacteria (GYEA)

The GYEA should serve as a special agar for sporulating bacteria like *Bacillus* and *Streptomyces*. The CFU/mL, after all PTW-treatment times and the TW treatment which were recorded at day 2 after treatment, were stable over the different days of incubation. The TVC_GYEA_controls_ were at a comparable level to TVC_MM_controls_. The TVC_GYEA_60s_ decreased (0.52 log_10_). The TVC_GYEA_240s_ dropped (1.65 log_10_), which was similar to the other agars. The TVC_GYEA_420s_ was reduced to the same level as TVC_GYEA_600s_ (3.23 log_10_ and 3.38 log_10_). However, TVC_GYEA_420s_ and TVC_GYEA_600s_ increased with longer incubation times. The TVC of the counterexample TVC_GYEA_600s_TW_ was reduced similar to TVC_GYEA240s_.

#### 3.1.4. Proliferation Assay of the Special Agars for Coliform Bacteria (KVGL and ENDO)

The agars KVGL and ENDO were both used as special agars to detect changes of the CFU/mL of coliform bacteria. KVGL was incubated at room temperature and ENDO at 37 °C. The TVC_KVGL_controls_, TVC_KVGL_60s_ (0.48 log_10_), TVC_KVGL_240s_ (1.63 log_10_), and TVC_KVGL_600s_TW_ (1.64 log_10_) underwent a stable reduction, until day 2, which were comparable to TVC_MCA_. A mild increase of CFU/mL over the increased incubation times was only detectable at longer PTW-treatment times.

The TVC_ENDO_ showed generally higher TVC at different PTW-treatment times compared to TVC_KVGL_. The TVC_ENDO_controls_ were a similar level to the TVC_KVGL_controls_. TVC_ENDO_60s_ was reduced likewise. The TVC_ENDO_240s_ was lowered (1.53 log_10_) compared to TVC_KVGL_60s_. The TVC_ENDO_420s_ and TVC_ENDO_600s_ had nearly the same level of reduction (3.40 log_10_ and 3.25 log_10_) and showed an increase of TVC until day 5 of incubation. In addition, TVC_KVGL_600s_TW_ and TVC_ENDO_600s_TW_ were nearly on the same level.

### 3.2. The Detected Live/Dead-Ratio after Different PTW-Treatment Times

The G/R-ratio was measured after different PTW-treatment times (Figure 4).

A reduction of the G/R of 32.9 ± 5.6% was achieved after 1 min of PTW treatment. With a PTW treatment of 4 min, the G/R ratio was reduced by 38.4 ± 4.0%. Longer treatments resulted in a G/R-ratio reduction of 46.8 ± 4.5% (7 min) and 49.2 ± 2.2% (10 min), respectively.

The strongest decrease in the G/R ratio compared to the control was observed after a 1 min PTW treatment (>30% reduction). With longer PTW-treatment times, the reduction efficiency continued to increase. However, the reduction with longer PTW-treatment times was not as efficient as the first interval (0–1 min).

### 3.3. Detection of Bacterial Groups Using MALDI-ToF Analysis

Isolates of different PTW-treatment times were used for MALDI-ToF analysis. Fifty-one isolates were sent for determination after repeated plating and incubation (PCA; room temperature).

The MALDI-ToF results (Figure 5) showed that *Pseudomonas* spp. comprised the majority of the TVC_PCA_. *Pseudomonas* spp. comprised approximately 37.0% of all hits. None of the *Pseudomonas* isolates could be analyzed down to species level. The second-largest group (19.6%) were the ‘unknown’ isolates that had no hits in the reference MALDI-ToF database. The third-biggest group was the *Bacillus* group (13.6%). This group was divided into the *Bacillus pumilus* group (5.8%) and *Bacillus cereus* group (7.8%). However, isolates of both *Bacillus* groups could not be analyzed down to the species level. 

In most of the PTW-treatment times, the genus of *Pseudomonas* was most detected. Only at the PTW-treatment time of 4 min was *Peribacillus* spp. detected at a higher percentage. Here, *Pseudomonas* followed with the second-highest detection percentage.

Isolates of controls included the two bacterial strains *Serratia plymuthica* and *Brevundimonas versicularis*, in addition to *Pseudomonas* spp. and some unidentified microorganisms. After one minute of PTW treatment, only *Pseudomonas* spp. was determinable. At 4 min of PTW treatment, the microorganisms of the *Bacillus cereus* group, *Flavobacterium* spp. and *Solibacrillus silvestris*, were also abundant on PCA.

After 7 min of treatment, the microorganisms *Priestia* spp., *Flavobacterium* spp., and *Solibacillus silverstris* were present, in addition to *Pseudomonas* spp. At the longest treatment time of 10 min, the two groups of *Bacillus* (*B. cereus* group and *B. pumilus* group) together were the second-most-dominant bacterial genus. Less frequently represented were the microorganisms *Microbacterium testaceum*, *Priestia* spp., and *Peribacillus* spp.

### 3.4. Characterization of the Anion and Cation Content of Lettuce after Washing with PTW or Tap Water with Ion Chromatography

Samples of PTW-treated and tap-water-treated lettuce were analyzed using ion chromatography to detect the concentrations of anions and cations (Figure 6).

Some of the anion and cation concentrations were influenced by the different washing techniques. On the side of the cations, concentrations of the sodium, potassium, magnesium, and calcium were reduced slightly after the PTW washing. Sodium was the tested cation with the highest concentration (~1900 mg/L), followed by potassium (~1600 mg/L), calcium (~550 mg/L), ammonium (~100 mg/L), and magnesium (~95 mg/L). This order has not been changed despite reduction. The PTW washing seems not to influence the ammonium concentration in the lettuce. The washing with conventional tap water showed nearly the same influence on the cation concentrations in the lettuce as the PTW washing.

In contrast to the cation household, the influence of the PTW and tap washing showed a different influence to the anion household. In descending order of concentration, carbonate (~4900 mg/L) was the most-concentrated anion, followed by sulfate (~2300 mg/L), nitrate and bromide, at nearly the same level (~1200 mg/L), chloride (1100 mg/L), and phosphate (~450 mg/L). Nitrite was not detectable. After the PTW washing, the concentration of carbonate was up to 10% higher than before. The concentrations of sulfate (−10%) and bromide (−25%) were lower after the PTW washing. The tap water washing resulted in a lowering of carbonate, bromide, sulfate, and phosphate.

## 4. Discussion

The goal of the presented work was to establish an innovative way for gentle fresh-cut-lettuce sanitation. Further studies showed that physical plasma has a reductive effect on microbial cells [19,30,31,33,38]. Our hypothesis was that PTW has a cell-wall-independent, non-selective reductive effect on microbial cells. The methods used confirmed that it was possible to show that all tested microbial groups were reduced in their TVC and viability. Our results suggest that treatment with PTW results in a uniform reduction of the microbial biomass, thus confirming our hypothesis regarding reduction. Interestingly, Gram-positive bacteria and fungi seem to recover better from treatment.

Proliferation assays were completed with different special agar types to investigate the impact of PTW on different cultivable microbial groups. Tests with full and minimal medium showed that the samples showed no significant differences within these microbial groups. It is always possible that some microorganisms may only grow under specific conditions which were not met in the experimental design. Lettuce contains a multispecies complexity of native microbial communities. In most cases, it is challenging to identify this with visual inspection, particularly if some microorganisms were replaced with others on the specific agar.

Therefore, the special agars for Gram-negative bacteria (MCA), Gram-positive bacteria (CNA), and yeasts and fungi (S4) were used to analyze the influence of PTW treatments on the three different microbial groups. The three tested microbial groups showed different susceptibilities to PTW. Gram-negative bacteria seem to be highly influenced by the PTW treatment and appear not to recover following it. In contrast to the large reductions in Gram-negative bacteria, the yeasts and fungi as well as the Gram-positive bacteria seem to cope with the PTW treatment.

The differences in the observed survival rate of the investigated microbial groups may be explained by their different resistance capacities against the antimicrobial active and reactive species, as well as the low pH of the PTW. The higher sensitivity of Gram-negative bacteria is likely based on their bacterial envelope structure. Gram-negative bacteria have a thin peptide glycan layer in the periplasmic space between the inner and outer membrane, which make the envelope resistant to physical and chemical stresses [41]. However, Chislett et al. showed that RNS i.e., HNO_2_ can react with and destroy Gram-negative envelopes, lysed cells, and it reacts with intracellular components [42]. Additionally, aerobic Gram-negative bacteria have evolved diverse defense strategies against oxidative stresses [43]. If the redox stress is overwhelming the defense capacities, RNOS could reach the intracellular space and may destroy the cellular machinery. Because of the high human health impact of coliform bacteria, these Gram-negative bacteria were tested further. However, the same dynamics for coliform bacteria were observed as for the total Gram-negative bacteria on MCA.

Gram-positive bacteria have a much thicker cell wall compared to Gram-negative [44]. This bacterial group has only one inner membrane but a strongly formed peptidoglycan layer [45]. This thick layer is thought to protect the cellular machinery under harsh conditions [46].

Therefore, it was unexpected to detect comparable reductions between bacteria on CNA (Gram-positive) and MCA (Gram-negative). However, increased incubation time of bacteria on CNA after PTW treatment resulted in increased TVC. One reason for this could be their ability to form endospores under non-optimal growth conditions. A large number of Gram-positive bacteria can also form endospores, with which they gain a high defense capacity against different stressors, including low water content and downregulated cell cycles [47]. Investigations by other scientists with microwave-driven physical plasma modified air (PPA) and *Bacillus athropheus* endospores resulted in a reduction of 3 log_10_ CFU/mL [48]. Therefore, sporicidal plasma reagents may reduce endospores as well as vegetative Gram-positive bacteria. On the other hand, survival of single cells may allow them to regrow under better conditions. These assumptions were also underlined by the obtained results of TVC on GYEA, which was used as special agar for sporulating bacteria like *Streptomyces* or *Bacillus*. TVC_GYEA_ showed nearly the same dynamics as TVC_CNA_ in the presented studies. Beside protective envelope structures and spore-forming capacity, acid-defense systems were described for Gram-positive bacteria. These may be efflux pumps, repair molecules, and alkali production [49,50]. Yeast and fungi also have strong envelopes with selective permeable barriers to protect them against environmental stresses [51]. In the presented research, the TVC_S4_ decreased with longer PTW-treatment times, which was in line with the results for the other investigated microbiological groups. In addition, comparable to TVC_CNA_ (Gram-positive bacteria), the TVC_S4_ increased with longer incubation times after the PTW treatment. However, this regrowth was not as strong as that detected for the Gram-positive bacteria. Another study with plasma gas as the antimicrobial sanitation method showed comparable reduction dynamics [52,53]. Nevertheless, the regrowth was notable and should not be neglected. Another point may be that yeasts and fungi have longer growth rates than bacteria, which could explain the later increase of the TVC_S4_ after longer incubation times. Longer regrowth times compared to bacterial cells after plasma treatments were also shown in the studies of Mann et al. (2015) [54].

With the usage of the LIVE/DEAD BacLight Bacterial Viability Kit, reductions of living cells were detectable based on cell envelope integrity. With longer PTW-treatment times, a lowering of intact cell envelopes was detected. However, the reduction of ≤50% was not as high as the reduction of the TVC (≥99.99% ≙ ≥4 log_10_). At first glance, these results seem to be contrary to each other. However, they may be an indication of a possible “viable but noncultivable” (VBNC) status in the case of bacteria [55].

VBNC means that microbial cells are still alive, but have temporarily reduced their metabolism to maintenance levels [56]. Stopping building their metabolism leads to the loss of their ability to grow on culture media [57,58]. Presumably, they were affected so strongly by PTW treatment (protein and membrane degradations) that the cells are primarily involved in the rebuilding of their machinery, which uses their lipid and carbohydrate metabolism as an energy source [59,60]. Hence why, while repairing, microorganisms do not start cell division and may appear as non-proliferative on the agar [59]. This may result in bias for cultivation-dependent investigations. To gather better insights on the metabolic activity of bacteria and other microorganisms, analyses based on metabolic activity like XTT assays or fluorescence in situ hybridization (FISH) assays should be conducted in future research.

The MALDI-ToF results showed a high detection rate of *Pseudomonas* spp. over all treatment times. Different strains of *Pseudomonas* have been described on plants [61,62]. Some of them are harmful and others beneficial for plant life [63,64]. However, a more specific differentiation of the single *Pseudomonas* spp. was not possible using MALDI-ToF analysis. Therefore, it could not be concluded that the PTW treatment would only kill the harmful pseudomonades. It is more likely that the PTW treatment would affect all pseudomonades in the same range. This could also be concluded for all microorganisms on the fresh-cut endive. Interestingly, members of *Bacillus* groups seem to be more resistant to longer PTW-treatment times than the *Pseudomonas* group. This is to be expected, due to the higher survival rates of Gram-positive bacteria in the performed special-agar-based experiments. Among the strains of *B. cereus* and *B. pumilus*, some human pathogens may occur. These strains can cause diarrhea by emitting heat-resistant enterotoxin [65,66]. Therefore, even after longer PTW-treatment times, the potential of human pathogen occurrence may be reduced but not completely removed. Another interesting result was the absence of fungi in the MALDI-ToF detections, which remained in contrary to the TVC results. Whether this result was a coincidence or based on the database of the MALDI-ToF cannot be answered clearly in retrospect.

Although coliform bacteria were detected on special agars, none could be detected using MALDI-ToF. It is likely that, as the amounts of coliform were reduced compared to other microbial groups on the PCA, they may not have been selected with the random picking of colonies. However, many studies showed that coliform could be present on ready-to-eat vegetables [67,68]. As coliforms are potential human pathogens, they are a bacterial group of special interest and have also been used as indicator organisms for fecal contamination of water and food [69]. Future investigations may be focused in more detail on human pathogens, including coliforms.

An important point of interest is the reduction of potentially harmful chemical compounds in foods. Some studies show potential health risks connected with nitrite/nitrate explosion [70,71]. In the gastrointestinal tract food-based nitrates could be formed into nitrites and nitrogen acids, which could react with amines forming potentially carcinogenic nitrosamines [72].

The tested MidiPLexc produced RNOS, which reduced the microbial load [33,73]. To be sure that no high potential for harmful nitrate or nitrite concentrations remained in the lettuce after the PTW-treatment cation and anion concentrations in the lettuce were tested. It seems that the cation and anion household were influenced slightly by the PTW-treatment. While no nitrite was detectable in all of the lettuce samples, the nitrate concentration (unwashed ~1250 mg/L) was higher after the PTW treatment (~1500 mg/L (+25%)). Interestingly, the nitrate concentration was increased, also, after the tap-water treatment (~1600 mg/L (+33%)). It seems that the increasing nitrate level was based more on the water utilized than on the plasma treatment. According to EU Regulation No. 1258/2011, the maximum values for nitrate for fresh lettuce from the open field are 4000 mg NO_3_/kg FW between October and March and 3000 mg NO_3_/kg FW between April and September (http://data.europa.eu/eli/reg/2023/915/2023-08-10 accessed on 1 December 2023). Overall, the nitrate concentrations of the treated lettuce are below the acceptable threshold of the EU regulations.

Another benefit of the PTW is that it has, aside from nitrate, no remaining disinfection byproducts like chlorides which made the wastewater more ecofriendly than, for example, chlorinated water [74].

## 5. Conclusions

The PTW treatment inactivated a wide spectrum of microorganisms on fresh-cut endive. The developed cultivable microbial spectrum of endive (Gram-positive and Gram-negative bacteria as well as fungi, molds, and spore formations) was as expected and is comparable to that in the existing literature. A clear reduction of TVC and the lethal effects on membrane integrity were achieved for all of the tested microbial groups. This widespread reduction efficiency could help to provide safe food for customers and could finally contribute to public health and disease prevention.

These microbial reductions were dependent on the microbes’ cultivability and known defense mechanisms (Gram-negative bacteria > molds/fungi > Gram-positive bacteria). However, as the experiments did not investigate the metabolic activity of lettuce-occurring microorganisms, VBNC status for bacteria could not be excluded. The possible changes in the metabolic activity of the surface microorganisms is a fascinating field of research for the future.

The non-selective or less-selective inactivation ability of PTW was unexpected. On the one hand, this could mean that it is difficult to separate plant-protecting commensals from plant and human pathogens. On the other hand, it indicates the broad application potential of the innovative plasma technology against microbial food spoilage. The development of microbial resistance and susceptibility, as observed with antibiotic substances, does not appear to occur, as the modes of action are broadly diversified.

## Figures and Tables

**Figure 1 foods-13-00282-f001:**
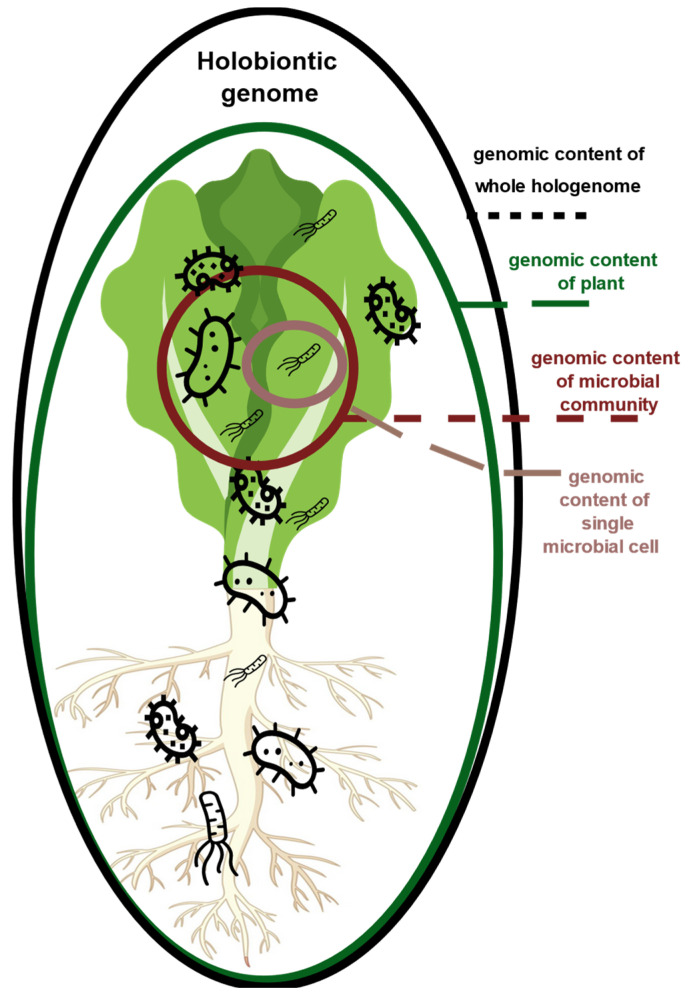
Illustration of possible holobiontic organisms.

**Figure 2 foods-13-00282-f002:**
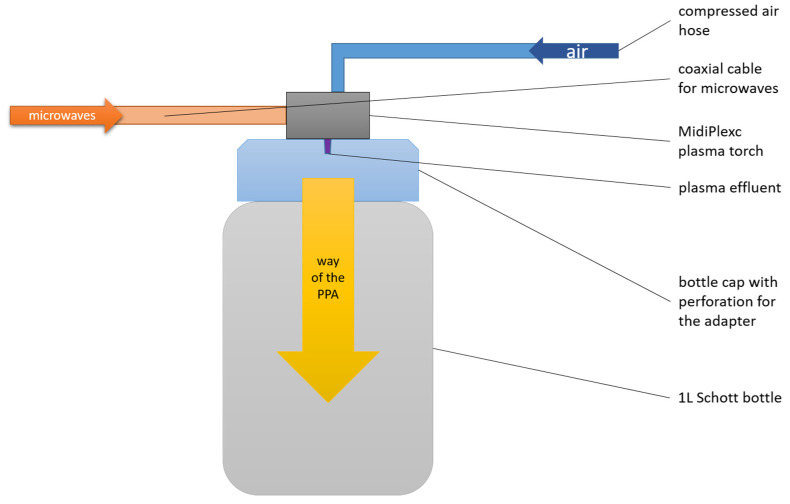
Structure of the plasma system.

**Figure 3 foods-13-00282-f003:**
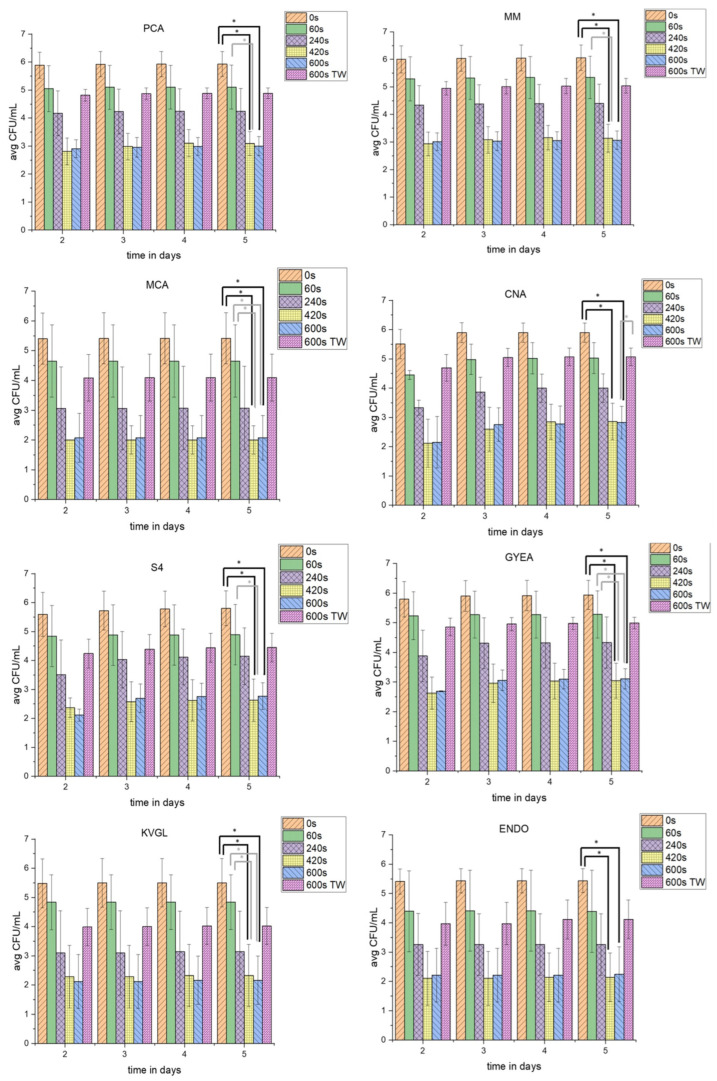
The influence of PTW treatment on different cultivable microbial groups. This figure shows the different TVC in CFU/mL of 5 g lettuce on different agars (PCA, MM, MCA, CNA, S4, ENDO, GYEA and KVGL) after different plasma-treatment times (0 s as control, 60 s, 240 s, 420 s, 600 s, and 600 with tap water (TW)) for native lettuce microbial communities. The plates were incubated for 5 days, and the TVC were counted on days 2–5. * depicts significant differences between two treatments (tested with Kruskal–Wallis and Dunn’s post hoc analysis, *p*-value < 0.0033).

**Figure 4 foods-13-00282-f004:**
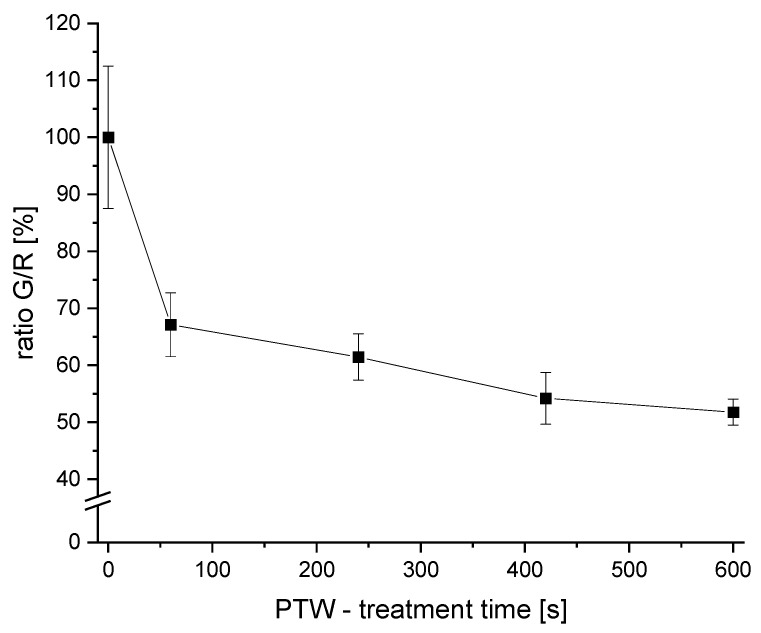
The G/R ratios of PTW-treated samples. Different PTW-treatment times were used (0 s as control, 60 s, 240 s, 420 s, and 600 s). The points of different treatment times are connected with a visual line.

**Figure 5 foods-13-00282-f005:**
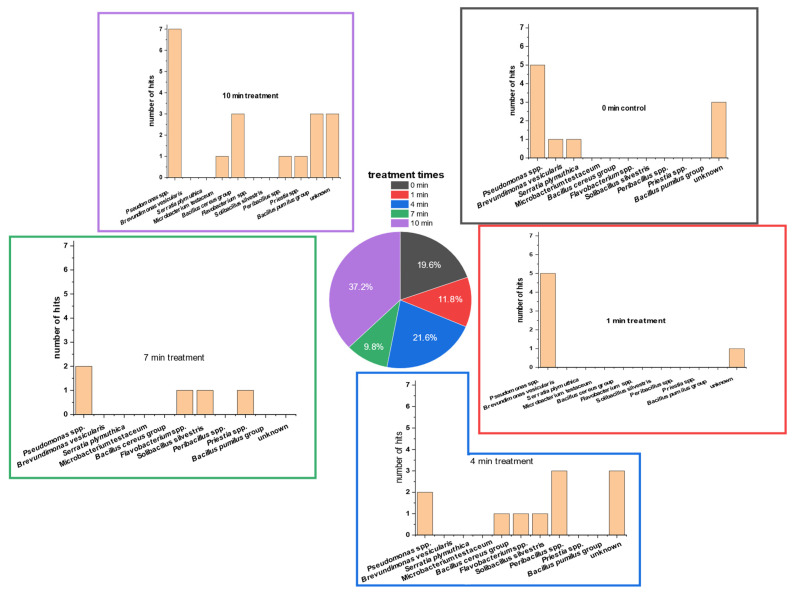
Different MALDI-ToF database hits of 51 randomly picked colonies of TVC (native lettuce microbial communities) on PCA of 5 g of lettuce treated with PTW at different treatment times (0 s as control, 60 s, 240 s, 420 s, 600 s). The plates were incubated for 5 days. After incubation, single colonies were picked randomly. These colonies were isolated on fresh PCA plates, transferred several times, and re-cultivated on fresh PCA. Finally, the isolates were sent for external analyses and identification of the colonies.Only five species of all isolates could be identified completely using MALDI-ToF analyses. All completely identified isolates were bacteria species: *Serratia plymuthica*, two strains of *Solibacillus silvestris, Micrpbacterium testaceum*, and *Brevundimonas vesicularis.* Further identified groups were *Priestia* spp., *Peribacillus* spp., and *Flavobacterium* spp.

**Figure 6 foods-13-00282-f006:**
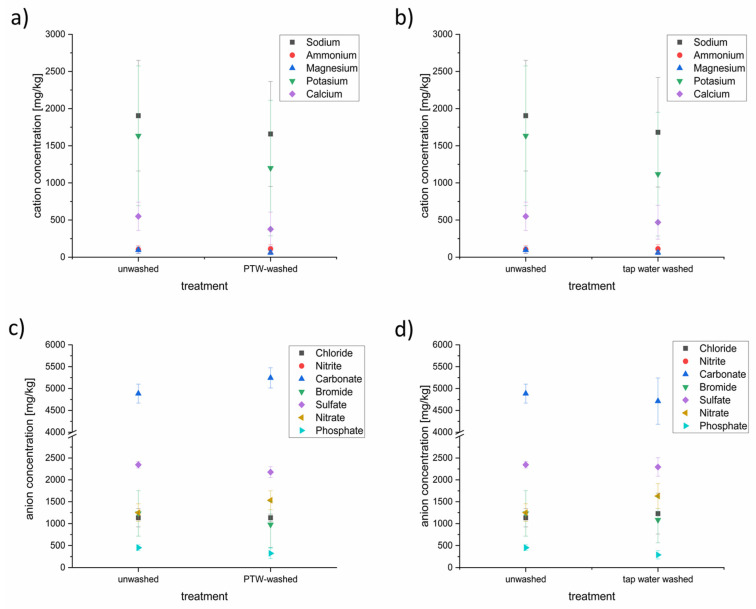
Cation (**a**,**b**) and anion (**c**,**d**) concentrations in PTW- and tap-water-treated lettuce samples. Depicted is the comparison of unwashed lettuce and PTW-washed lettuce (**a**,**c**) as well as the comparison of unwashed lettuce against tap-water washed lettuce (**b**,**d**).

**Table 1 foods-13-00282-t001:** Compositions of the different agars.

Agar Type	Cultivable Microbial Groups	Agar Composition	Company
Plate count agar (PCA)	Full-medium agar	23.5 g/L	Carl Roth (Karlsruhe, Germany)
Standard count agar (MM)	Minimal-medium agar	25 g/L	Carl Roth (Germany)
MacConkey agar (MCA)	Special agar for Gram-negative bacteria	50 g/L	Carl Roth (Germany)
PCA with collistinesulfate and nalidixinacid (CNA)	Special agar for Gram-positive bacteria	PCA After autoclaving: 10 mg/L collistinsulfat 15 mg/L nalidixinacid	PCA; Carl Roth (Germany) Collistinsulfat; Carl Roth (Germany) nalidixinacid; Carl Roth (Germany)
Sabouraud agar with 4% dextrose (S4)	Special agar for yeasts and molds	65 g/L	Merck KGaA (Darmstadt, Germany)
Glycerol yeast extract agar (GYEA)	Special agar composition for sporulating bacteria	15 g/L agar-agar 5 g/L glycerol 2 g/L yeast extract 1 g/L di-potassium phosphate	Agar-argar, Kobe I; Carl Roth (Germany) Glycerol; TH Geyer (Renningen, Germany) Yeast extract; Carl Roth (Germany) Di-potassium-phosphate; Carl Roth (Germany)
Crystal violet bile lactose agar (CVBLA)	Special agar for coliforms (room temperature)	41.5 g/L	Carl Roth (Germany)
Endo agar (ENDO)	Special agar for coliforms (37 °C)	36 g/L Endo agar base agar 5 mL/L ethanolic fuchsin solution	Endo agar base, Carl Roth (Germany) Fuchsin, Carl Roth (Germany)

## Data Availability

The data presented in this study are available on request from the corresponding author. The data are not publicly available due to ongoing work.

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
