# Peer review of "Deep Impact: Shifts of Native Cultivable Microbial Communities on Fresh Lettuce after Treatment with Plasma-Treated Water"

_foods, 2024, doi:10.3390/foods13020282_

Round 1
Reviewer 1 Report
Comments and Suggestions for Authors
Manuscript ID: foods-2762359
Reviewer comments:
1. The logic of the introduction is quite confusing, it is not clear why PTW is used, what is the purpose of the research in this paper? Here, we propose to rethink the logic and improve the clarity of linguistic expression.
2. Did the authors consider the cost of PTW, whether it is suitable for the public or the factory to use the method?
3. Abstract needs to be modified.
4. Figure 4 does not reflect the significance analysis, it is suggested to add this part to improve the quality of the figure.
5. Please add the number of microorganisms detected in the postharvest samples and compare them with the samples taken at the end of refrigeration to see if refrigeration has any effect on them.
6. It is recommended to add the availability of parallel experiments for all experiments in detail in the method description.
7. The Discussion Section is still a description of the phenomenon without in-depth discussion of the results or comparative analysis with others, so it is suggested to reorganize the language to enhance the scientific nature of the section.
8. It is recommended to clarify the purpose and significance of the study when writing the Conclusion.
9. There has been lack of references in the last three years, so it is recommended to add more recent references.
There seems to be no obvious grammar mistakes in this paper.
Author Response
Dear reviewer,
thank you for your suggestions.
We have tried to incorporate your comments as good as possible.
1. The logic of the introduction is quite confusing, it is not clear why PTW is used, what is the purpose of the research in this paper? Here, we propose to rethink the logic and improve the clarity of linguistic expression.
Thank you very much for the hint. The introduction was shorted and in parts re-written.
2. Did the authors consider the cost of PTW, whether it is suitable for the public or the factory to use the method?
Thank you for this question. Yes, we considered the costs of PTW for an up-scaled version of the used plasma source (please see reference: https://doi.org/10.1007/s12393-020-09238-9 Food Engineering Reviews (2021) 13:115–135). As we are able to produce 60 L of PTW within 1 h for the moment, we have 0.08 €/L production costs of undiluted PTW based on current costs for water and power in Germany in 2023. In other countries it can be much cheaper. Currently, in Germany, it is the double price you need to pay for chlorine dioxide application or the same like you pay for peracetic acid and the fourth of just using water without additions. We work in all projects together with the industry and related to them the price is acceptable for the additional benefit.
Abstract needs to be modified.
Thanks for the hint. It is now modified.
4. Figure 4 does not reflect the significance analysis, it is suggested to add this part to improve the quality of the figure.
Thank you very much. No statistical testing was performed, since the confidence overlap. Overlapping confidence intervals could be an identifier for no/weak significance. Therefore, the testing was omitted to prevent misleading results. For better statistical testing conditions, more independent experiments would be necessary.
5. Please add the number of microorganisms detected in the postharvest samples and compare them with the samples taken at the end of refrigeration to see if refrigeration has any effect on them.
Thank you for the good hint. The tested lettuce was handled like in the industry. The lettuce heads were sampled by the harvest helper. So, we could not test the TVC before the cooled delivery. But we will try to find a way in the next sampling campaign.
6. It is recommended to add the availability of parallel experiments for all experiments in detail in the method description.
Thank you very much for the hint. The method part is now extended.
7. The Discussion Section is still a description of the phenomenon without in-depth discussion of the results or comparative analysis with others, so it is suggested to reorganize the language to enhance the scientific nature of the section.
Thanks for the hint. The discussion has been revised.
8. It is recommended to clarify the purpose and significance of the study when writing the Conclusion.
Thanks for the hint. The conclusion part has been modified.
9. There has been lack of references in the last three years, so it is recommended to add more recent references.
Thank you very much for the note to use more recent references. We added them.
Comments on the Quality of English Language
There seems to be no obvious grammar mistakes in this paper.
Reviewer 2 Report
Comments and Suggestions for Authors
The paper, titled 'Deep Impact: Shifts of Native Cultivable Microbial Communities on Fresh Lettuce after Treatment with Plasma-Treated Water', discusses the use of plasma-treated water to eliminate or reduce microorganisms in food. The authors present an innovative and alternative method of food sanitation using atmospheric pressure thermal plasma.
However, doubts were raised during the introduction stage. The abstract states that 'One innovative sanitation method is non-thermal plasma, which offers an extremely efficient reduction of microbial live.' Meanwhile, in the introduction, it is suggested that 'An innovative and alternative method for food sanitation may be the use of non-thermal atmospheric pressure plasma'.
Overall, the paper's subject matter is intriguing, but requires further organization. Suggestions and comments will be provided in subsequent points.
1. The introduction is excessively prolonged and disorganised. In my view, the various types of plasma utilised and the techniques for producing plasma water ought to be examined.
2. The method elucidated appears contentious to me since the authors describe in their paper that the impact of plasma may result in the creation of nitrogen oxides, nitric acid, among other substances. Overall, there is a growing trend towards reducing nitrates in food products and minimising the use of nitrate fertilisers due to the potential formation of harmful compounds resulting from reactions with nitrogen oxides, such as nitroso-derivatives. The authors should provide an explanation for this issue.
3. Additionally, the test material lacks proper characterisation, including measurements of pH, conductivity, nitrogen oxide content, oxygen, sulphur, and other relevant compounds. This work requires an analysis to establish links between the mechanism of interaction with microorganisms or fungi.
4. It would be more worthwhile if the authors conducted a thorough investigation of the literature and presented the pros and cons of applying plasma water.
The research material requires thorough examination, substantial revision, and verification of the absence of any product interactions. For instance, the nitrate content of lettuce should be analysed before and after product interaction.
Author Response
Dear reviewer,
thank you for your suggestions.
We have tried to incorporate your comments as good as possible.
Comments and Suggestions for Authors
The paper, titled 'Deep Impact: Shifts of Native Cultivable Microbial Communities on Fresh Lettuce after Treatment with Plasma-Treated Water', discusses the use of plasma-treated water to eliminate or reduce microorganisms in food. The authors present an innovative and alternative method of food sanitation using atmospheric pressure thermal plasma.
However, doubts were raised during the introduction stage. The abstract states that 'One innovative sanitation method is non-thermal plasma, which offers an extremely efficient reduction of microbial live.' Meanwhile, in the introduction, it is suggested that 'An innovative and alternative method for food sanitation may be the use of non-thermal atmospheric pressure plasma'.
Overall, the paper's subject matter is intriguing, but requires further organization. Suggestions and comments will be provided in subsequent points.
- The introduction is excessively prolonged and disorganised. In my view, the various types of plasma utilised and the techniques for producing plasma water ought to be examined.
Thanks for the hint, the introduction was reduced.
- The method elucidated appears contentious to me since the authors describe in their paper that the impact of plasma may result in the creation of nitrogen oxides, nitric acid, among other substances. Overall, there is a growing trend towards reducing nitrates in food products and minimising the use of nitrate fertilisers due to the potential formation of harmful compounds resulting from reactions with nitrogen oxides, such as nitroso-derivatives. The authors should provide an explanation for this issue.
Thank you very much for this comment. You are right, nitrate fertilizers should be reduced in the field and nitrates in food should also be decreased, especially for meat and cheese. As the used PTW contains specific amounts of nitrite and nitrate antimicrobial effects are achieved. On the other hand, fresh-cut lettuce already contains nitrate in higher concentrations (3000-4000 mg NO3-/kg FW are within accepted thresholds, depending on Winter or Summer season). In addition, little amounts of nitrate can already be inside the clean water used for PTW generation. Therefore, we measured the nitrite and nitrate concentration of the lettuce before and after treatment within the upscaled version of the used plasma source (for the plasma source, please see reference: please see reference: https://doi.org/10.1007/s12393-020-09238-9 Food Engineering Reviews (2021) 13:115–135). The results for the nitrite and nitrate concentration of the fresh-cut lettuce before and after treatment is now given in the results part. Further other anions and cations are also given.
- Additionally, the test material lacks proper characterisation, including measurements of pH, conductivity, nitrogen oxide content, oxygen, sulphur, and other relevant compounds. This work requires an analysis to establish links between the mechanism of interaction with microorganisms or fungi.
Please see our answer for question number 2.
- It would be more worthwhile if the authors conducted a thorough investigation of the literature and presented the pros and cons of applying plasma water.
The research material requires thorough examination, substantial revision, and verification of the absence of any product interactions. For instance, the nitrate content of lettuce should be analysed before and after product interaction.
Thanks for the good hint. The content was now analysed.
Round 2
Reviewer 2 Report
Comments and Suggestions for Authors
The authors made many changes and responded to comments and questions. I believe that in this form the work is already much better and, in my opinion, is suitable for publication.